# The Complex Interplay between Immunonutrition, Mast Cells, and Histamine Signaling in COVID-19

**DOI:** 10.3390/nu13103458

**Published:** 2021-09-29

**Authors:** Sotirios Kakavas, Dimitrios Karayiannis, Zafeiria Mastora

**Affiliations:** 1Critical Care Department, “Sotiria” General Hospital of Chest Diseases, 152 Mesogeion Avenue, 11527 Athens, Greece; sotikaka@yahoo.com; 2Department of Clinical Nutrition, Evangelismos General Hospital of Athens, Ypsilantou 45-47, 10676 Athens, Greece; 3First Department of Critical Care Medicine and Pulmonary Services, Evangelismos General Hospital, National and Kapodistrian University of Athens, 11527 Athens, Greece; zafimast@yahoo.gr

**Keywords:** immunonutrition, COVID-19, histamine

## Abstract

There is an ongoing need for new therapeutic modalities against SARS-CoV-2 infection. Mast cell histamine has been implicated in the pathophysiology of COVID-19 as a regulator of proinflammatory, fibrotic, and thrombogenic processes. Consequently, mast cell histamine and its receptors represent promising pharmacological targets. At the same time, nutritional modulation of immune system function has been proposed and is being investigated for the prevention of COVID-19 or as an adjunctive strategy combined with conventional therapy. Several studies indicate that several immunonutrients can regulate mast cell activity to reduce the de novo synthesis and/or release of histamine and other mediators that are considered to mediate, at least in part, the complex pathophysiology present in COVID-19. This review summarizes the effects on mast cell histamine of common immunonutrients that have been investigated for use in COVID-19.

## 1. Introduction

Severe acute respiratory syndrome coronavirus 2 (SARS-CoV-2) is an enveloped single-stranded positive-sense ribonucleic acid (RNA) virus that was first detected in China and has caused an ongoing global pandemic [1]. SARS-CoV-2 comprises four identified structural proteins, namely, spike (S), membrane (M), envelope (E), and nucleocapsid (N) [2]. In general, the virus infects by binding its S protein to the host’s angiotensin-converting enzyme 2 (ACE2) receptors, then entering by endocytosis into airway epithelium cells, lung macrophages, alveolar epithelial cells, and vascular endothelial cells [3,4]. Patients may remain asymptomatic or develop symptoms of varying severity [5,6]. In the resulting coronavirus disease 2019 (COVID-19), activation of the innate immunity, specific antibodies, and activated T cells represent basic defensive factors, while in more severe cases, lung injury progresses and leads to respiratory failure [5,7]. Severe lung injury in SARS-CoV-2 patients is considered the result of immune hyperreaction that involves both innate and adaptive immune responses [6,8]. Briefly, coronavirus infection activates antigen-presenting cells, such as macrophages, that display viral antigens to T and B cells resulting in antibody production and increased cytokine secretion in the form of a cytokine storm. Other immune cells are also implicated, including mast cells, which are important coordinators for both innate and adaptive immunity [9]. Endothelial injury and microthromboses ensue in the lungs and other organs of COVID-19 patients [10,11]. Patients may require mechanical ventilation and develop multiple organ failure [5,6].

Histamine is an endogenous biogenic amine that functions as a neurotransmitter and an immunoregulatory factor. In the immune system, histamine is mainly stored in cytoplasmic granules of mast cells and basophils and is released upon triggering along with other mediators such as serotonin, proteases (e.g., tryptase and chymase), heparin, a variety of cytokines, and angiogenic factors [12]. Histamine release can be activated by numerous innate signals or exogenous triggers [13] including allergens, toxins, and viruses [14]. The high-affinity immunoglobulin (Ig)E receptor, FcepsilonRI (FcεRI), is the primary receptor in mast cells that mediates IgE-dependent (allergic) reactions [12]. Yet, it is apparent that non-IgE-mediated mechanisms of mast cell activation also exist [13]. Histamine exerts its biological actions through four types of G protein-coupled histamine receptors (i.e., H1 receptor, H2 receptor, H3 receptor, and H4 receptor) [15]. It also activates acute immune-mediated reactions and enhances vascular smooth muscle contraction and the migration of other immune cells, antibodies, and mediators to the site of insult [7]. The release of histamine by perivascular mast cells may also affect adjacent lymphatic vessel function inducing immune cell trafficking through its lumen, which potentially contribute to acute inflammatory stimulus [16]. In the lungs, this may cause bronchoconstriction, increased mucus production, increased vasopermeability with edema, microthrombosis, and infiltration by leukocytes, predominantly neutrophils [17]. Histamine can regulate the balance between Th1 and Th2 effector cells [18]. During histamine-mediated lung inflammation, secretion of Th2 cytokines is enhanced, while production of Th1 cytokines is suppressed [19]. This response may increase susceptibility to viral and bacterial infections of the respiratory tract [5]. In addition, viral remnants may prolong and exaggerate the inflammatory process, causing a histamine-induced release of more pro-inflammatory Th2 cytokines through an IgE-mediated positive feedback vicious cycle [5].

A growing body of evidence has implicated histamine and mast cells in COVID-19 [20,21,22]. In animal models of COVID-19, mast cells detected in the lungs were chymase positive [23]. Mast cells are shown to express histamine receptors by themselves which, in an autocrine fashion, can potentially ensue a feedback regulation further enhancing inflammatory response [16,24]. The SARS-CoV-2 infection has been shown to activate mast cells leading to histamine release that increases IL-1 levels, causing hyper-inflammation and cytokine storm [25]. Mast cell degranulation has been reported in alveolar septa of deceased patients with COVID-19 and in SARS-CoV-2-infected mice and non-human primates [23,26]. Furthermore, this mast cell activation was associated with interstitial edema and immunothrombosis [27], while the levels of the mast cell-specific protease, chymase, correlated significantly with disease severity [23]. Moreover, studies have reported that H1 as well as H2 receptor antagonists, such as famotidine, are associated with a reduced risk of infection and deterioration leading to intubation or death from COVID-19 [28,29]. These agents are considered to improve pulmonary symptoms of SARS-CoV-2 infection by blocking the histamine-mediated cytokine storm [30]. Nevertheless, these observational findings need further validation by the ongoing randomized clinical trials.

Given that limited therapeutic modalities are available for the treatment of COVID-19, nutritional modulation of the immune system function has been proposed and is being investigated [31,32,33,34]. It is widely accepted that normal nutritional status is vital for immune homeostasis [35], while a number of recently published key studies suggest promising effects of immunonutrition on acute respiratory infections [36,37]. Briefly, immunonutrition can be defined as modulation of either the activity of the immune system or modulation of the consequences of activation of the immune system by nutrients or specific food items fed in amounts above those normally encountered in the diet [38]. Until now, specific immunonutrients have been proposed as effective for the prevention of COVID-19 or as an adjunctive strategy combined with conventional therapy [39]. At the same time, these nutraceuticals have been reported to modulate mast cell activation and histamine release with similar potency to pharmacological interventions [40,41]. This review summarizes the effects on mast cell and histamine signaling of common immunonutrients that have been investigated for use in COVID-19.

## 2. Vitamins

### 2.1. Vitamin D

Vitamin D has been linked to the susceptibility to SARS-CoV-2 infection and the prognosis of COVID-19 based on a series of data [32]. There is evidence that vitamin D inhibits the entry and replication of SARS-CoV-2 and suppresses the levels of pro-inflammatory cytokines while enhancing the production of anti-inflammatory cytokines and antimicrobial peptides [42]. According to epidemiological observations, vitamin D deficiency has been associated with a higher risk, severity, and mortality rate of COVID-19 [43,44]. However, conflicting results have been reported concerning the effects of vitamin D supplementation in outpatients and hospitalized patients after COVID-19 diagnosis in terms of disease severity, hospital length of stay, ICU admission, or mortality rate [45,46,47,48]. Although, no official guidelines exist, it has been proposed to aim for adequate serum 25(OH)D levels of at least 30 ng/mL (75 nmol/L) during the pandemic [49]. Further results are pending ongoing clinical trials [50].

Vitamin D seems to preserve the stability of mast cells, possibly by maintaining the expression of vitamin D receptors. In a vitamin D-deficient environment, mast cell activation occurs automatically, even in the absence of specific triggering [51]. In addition, it has been shown that vitamin D inhibits histamine release from mast cell activation including IgE-mediated activation [52]. Likewise, decreased levels of serum histamine have been found after the antigenic challenging of sensitized mice previously receiving a vitamin D supplemented diet [51]. According to this study, vitamin D receptor binding inhibits mast cell activation by blocking the non-receptor tyrosine kinase Lyn. Lyn is recruited immediately during mast cell activation following the crosslinking of FcεRI–IgE complexes by multivalent antigens or exposure to the bacterial lipopolysaccharide [53,54]. Furthermore, the phosphorylation of the Syk tyrosine kinase was also suppressed by vitamin D receptor binding to the β chain of FcεRI. Syk activation can be triggered by Lyn and is involved in mast cell degranulation [55]. Recent data also indicate a positive effect of vitamin D supplementation on functional humoral immunity levels as determined by IgG levels [56].

### 2.2. Vitamin E

Vitamin E is a lipid-soluble vitamin with antioxidant and immunomodulatory properties. In addition to scavenging free radicals, vitamin E can affect immune function by modulating signal transduction and gene expression [57,58,59]. In this way, vitamin E has been found to reduce susceptibility to respiratory infections as well as allergy-related diseases such as asthma [59]. Vitamin E has been implicated in the treatment of SARS-CoV-2 infection in an effort to minimize oxidative damage in these patients [33]. However, limited evidence exists on the use of vitamin E as an adjuvant agent for the treatment of COVID-19 patients, and information resulting from clinical trials is wanted [60]. 

Vitamin E has been shown to have an inhibitory effect on the proliferation, secretion, and survival of mast cells [61]. This effect originates from the modulation of protein kinase C, protein phosphatase 2A, and protein kinase B in mast cells. Furthermore, in vitro studies in various mast cell lines have shown that vitamin E affects mast cell activation, resulting in a decreased release of proinflammatory mediators including histamine [62,63]. The effects of vitamin E on mast cell function could be related with the antioxidative properties of the vitamin [61]. Interestingly, oxidative stress and mast cells interact and participate in acute lung injury. Reactive oxygen species generation promotes pulmonary mast cell degranulation which, in turn, can increase oxidative stress and inflammation during acute lung injury [64]. 

### 2.3. Vitamin C

Vitamin C or ascorbic acid is a water-soluble antioxidant vitamin that possesses anti-inflammatory and immunomodulatory properties [5]. Although the value of vitamin C has not yet been demonstrated in COVID-19, it has gained interest in this context because of its antiviral action [65] and beneficial effects in oxidative damage and inflammation [66]. Vitamin C has previously been implicated in sepsis and ARDS, both of which represent major complications of COVID-19 [67]. Although low levels of vitamin C have been reported in sepsis, conflicting results have been produced by studies evaluating vitamin C supplementation in septic shock and ARDS [68,69]. At present, we are awaiting the results of several ongoing trials evaluating the value of oral or intravenous vitamin C supplementation in the treatment of COVID-19. A daily oral dosage of 1–2 g/day of vitamin C has been proposed as beneficial for the prevention or treatment of COVID-19, while higher doses of intravenous vitamin C, up to 24 gm/day, are being evaluated in critically ill patients with COVID-19. Proposed mechanisms for the ability of vitamin C to benefit patients with COVID-19 point to the prevention of IL-6 increase in several (pro)inflammatory conditions and the inhibition of increases for a range of inflammatory cytokines [70,71].

Previous studies have shown that vitamin C administration attenuates a robust immune response [72]. In fact, mast cell-mediated bronchial hypersensitivity caused by the common cold was inhibited by the administration of vitamin C [73]. These patients exhibited decreased bronchial hypersensitivity to histamine and bronchoconstriction after vitamin C administration [40]. Both preclinical [74,75,76] and clinical studies [76,77,78] have evaluated histamine blood levels after vitamin C administration. In a recent study, 7.5 g of vitamin C administered intravenously in 89 patients with allergies or upper respiratory infections caused a significant reduction in serum histamine [79]. Several mechanisms may be responsible for the inhibitory effect of vitamin C on histamine [79,80]; vitamin C may inhibit mast cell activation, increase histamine degradation by diamine oxidase or, alternatively, decrease histamine production by inhibiting histidine decarboxylase [81].

## 3. Minerals

### 3.1. Zinc

Zinc is the second most abundant essential trace element that plays important roles in the development, differentiation, and function of immune cells [33]. The perceived antiviral properties of zinc against upper respiratory tract viral infections derive from its participation in metallothioneins [82]. In this context, zinc may interfere with viral infection in many ways [83,84]. First, zinc may prevent viral attachment to nasopharyngeal mucous as well as fusion with the host’s membrane and virus entry into cells. In particular, zinc has been shown to decrease the activity of the ACE2 receptor, which is essential for SARS-CoV-2 binding and the provocation of cytokine storm. Moreover, this trace element has been shown to hinder SARS-CoV-1 viral replication by inhibiting SARS-CoV RNA polymerase [85]. Further antiviral effects of zinc include the impairment of viral protein translation and the blockade of viral particle release [86]. Zinc deficiency is common in COVID-19 patients and is associated with more complications and increased mortality [42]. In older adults, supplementation with 45 mg elemental zinc per day has been shown to reduce the risk of infection [31]. In summary, it has been proposed that zinc supplementation may be beneficial in the prevention and treatment of SARS-CoV-2 infection and the associated inflammation [87,88,89]. At present, a series of clinical trials have been registered to test the efficacy of various regimens containing zinc against COVID-19. 

Zinc deficiency has been demonstrated to affect the function of various types of immune cells including mast cells [90,91]. Zinc seems to be essential for mast cell activation. In an in vitro study, the release of histamine from human basophils and lung mast cells was inhibited from physiological concentrations of zinc [92]. A possible mechanism may include the blockade of Ca2+ influx induced by the IgE-mediated activation of mast cells [6]. On the other hand, a zinc chelator (N,N,N,N-tetrakis (2-pyridylmethyl) ethylenediamine) has been recently shown to contribute to the inhibition of histamine release from mast cells and this effect was reversed by zinc supplementation [93]. Zinc may regulate mast cell activation and function by modulating the PKC/NF-κB signaling pathway [90]. Various mechanisms have been suggested, but modulation of the NF-kB pathway could be the result of the inhibition of cyclic nucleotide phosphodiesterase, cross activation of protein kinase A, and inhibitory phosphorylation of protein kinase Raf-1 [94]. In addition, activation of NF-kB can also activate mast cells thereby releasing histamine secretion and an ensuing inflammatory response along with cytokine secretion [95].

### 3.2. Selenium

Selenium is a trace element that serves as an essential component of antioxidant enzymes. In this way, it exhibits a protective effect against respiratory infections including viral infections [96,97] (33, 97, 98). It has been suggested that selenium deficiency might be implicated in the evolution of SARS-CoV-2 [87]. Moreover, a number of studies have linked selenium with SARS-CoV2 infection and recovery rates [98,99,100]. Selenium may halt oxidative stress in patients with COVID-19 [33,39]. Interestingly, oxidative stress and mast cells show a bidirectional interaction. Intracellular reactive oxygen species production is the result of mast cells by various triggers [101], while mast cell degranulation can be controlled via the decrease in reactive oxygen species generation using antioxidants [62]. In accordance with this, an in vitro study showed that selenium can suppress the IgE-mediated release of inflammatory mediators in a murine mast cell line, although histamine release only slightly decreased [102]. The regulation of redox-sensitive transcription factors is considered the responsible mechanism by which selenium affects mast cell histamine release [103]. Published data also highlight the important role of biological functions that occur via incorporation of selenium into selenoproteins in the form of selenocysteine amino acid residue. Selenocysteine (Sec-Cys) is involved in a variety of prostanoid metabolism processes and, therefore, have an impact on immunity [104].

## 4. Omega-3 Fatty Acids

Omega-3 fatty acids are polyunsaturated fatty acids (PUFAs) obtained mainly from two dietary sources: marine and plant oils. These fatty acids incorporate into the bi-phospholipid layer of the cell membrane and result in the reduced production of pro-inflammatory mediators [105]. To date, sparse evidence has implicated omega-3 fatty acids in the prevention and treatment of COVID-19 [106,107]. Nevertheless, it has been shown that the omega-3 PUFAs inactivate enveloped viruses like SARS-CoV2 and inhibit ACE2-mediated binding and cellular entry of SARS-CoV-2 [108]. Furthermore, beneficial reports of omega-3 PUFAs have been reported in patients with sepsis and sepsis-induced ARDS [109,110]. Several clinical trials assessing the effect of omega-3 PUFAs in COVID-19 management are currently registered (ZPD37). In a recent double-blind, randomized clinical trial, enteral supplementation with omega-3 PUFAs significantly improved respiratory and renal function indices as well as one-month survival rates in critically ill patients with COVID-19 [111].

Similar to other immune cells, fatty acids are incorporated into mast cell membranes and can differentially influence mast cell secretive properties [62,112,113]. Collectively, the actions of omega-3 PUFAs on mast cells are mainly inhibitory. A series of studies in animal models and in human cells has demonstrated the inhibitory effect of omega-3 PUFAs on IgE-mediated activation of mast cells [26,114,115]. This effect is mediated by the inhibition of GATA transcription factors in mast cells and leads to suppressed Th2 cytokine expression [116]. As expected, this action of omega-3 PUFAs was tested to ameliorate the severity of mast cell-associated diseases [117,118]. In a canine atopic dermatitis model mast cell histamine release was reduced after treatment by γ-linolenic acid or α-linolenic acid. On the other hand, linoleic acid or arachidonic acid enhanced histamine release [113,119]. However, in a model of stress-induced visceral hypersensitivity in maternally separated rats, neither mast cell degranulation nor hypersensitivity were affected by the administration of an omega-3 PUFA-enriched diet [120]. Clinical trials of the dietary omega-3 supplementation in asthma patients have reported beneficial effects on airway inflammation but inconsistent clinical benefits in terms of lung function indices [121]. Nonetheless, it should be noted that two of these studies reported clinical benefits of dietary supplementation with omega-3 PUFAs in asthma patients without an accompanying decrease in mast cell activation and histamine release [122,123].

## 5. Phytochemicals

### 5.1. Flavonoids

Flavonoids are a group of naturally occurring polyphenolic substances with anti-oxidative and anti-inflammatory actions in various disease states [124]. They may also have antiviral properties and several representatives of this family, such as quercetin, have been proposed as a potential treatment of COVID-19 [125,126]. Luteolin from Veronica linariifolia may also be beneficial, since it has been shown to prevent viral entry into the host cell by inhibiting the binding of the SARS-CoV spike protein [127]. A potential antiviral activity via the inhibition of the SARS-CoV helicase has been reported for luteolin, myricetin (from Myricanagi), and scutellarin (from Scutellaria barbata) [128]. Finally, the antiviral activity of kaempferol has been suggested to derive from the inhibition of the 3a-channel protein of SARS-CoV [129].

Several flavonoids inhibit in vitro the expression and/or release of mediators, such as histamine, by human and rodent mast cells [130,131,132]. More specifically, quercetin inhibits mast cell activation and release of histamine and may modulate airway inflammation [133,134]. Likewise, luteolin or a structural analog of luteolin inhibit mast cell activation and histamine release from animal and human mast cells [135,136,137]. The modulatory action of flavonoids on mast cell secretory function affects both IgE-dependent and independent processes and appears to be selective [130]. Some flavonoids, such as caffeic acid, inhibit selective histamine release, while others, such as luteolin and myricetin, inhibit both histamine and β-hexosaminidase release [138]. This inhibitory action may involve the suppression of NF-κB activation [137,139]. The inhibition of calcium influx and protein kinase C translocation and activity mediate the actions of luteolin and quercetin on histamine release from murine bone marrow-derived mast cells, rat peritoneal mast cells, and human cultured cord blood-derived mast cells [131,140,141]. Similarly, quercetin, kaempferol, and myricetin suppressed IgE-mediated activation and histamine release from human umbilical cord blood-derived cultured mast cells. The proposed mechanism includes the decrease of intracellular calcium influx and the inhibition of protein kinase C-theta isoenzyme signaling [140]. Finally, luteolin inhibits neuropeptide (non-IgE mediated) stimulation of mast cells via the mammalian target of rapamycin (mTOR) signaling [142].

### 5.2. Curcumin

Curcumin is a natural yellow constituent of turmeric or curry powder that is derived from the rhizome of Curcuma longa plants [143]. Curcumin has been reported as a pleiotropic molecule with various biological actions including antioxidant and anti-inflammatory effects [144]. The oral or intranasal administration of curcumin has been shown to suppress airway inflammation and remodeling and to inhibit airway hyperreactivity to histamine and bronchoconstriction in animal models of asthma [145,146]. Curcumin may also exhibit antiviral activities and has been shown to hamper the replication and proliferation of SARS-CoV-1, the first beta-coronavirus that caused the 2003 SARS outbreak and shares a substantial genetic similarity with SARS-CoV-2 [147]. Moreover, in a rat experimental model, curcumin administration resulted in the attenuation of myocardial fibrosis by modulating angiotensin receptors and ACE2 [4,148]. A similar role could be proposed in the fibrotic process that emerges as a secondary event in severe COVID-19 [148]. Along with its well-known anti-inflammatory effects, curcumin has been reported to inhibit mast cell degranulation and histamine release in vitro and in vivo [149,150,151]. A possible mechanism may include the in vivo suppression of the Syk-dependent phosphorylations, which are critical for mast cell activation. Although the phosphorylation of Syk itself was not affected, curcumin directly inhibited Syk kinase activity in vitro [149]. Curcumin also inhibited the phosphorylation of additional down-stream signaling molecules including Akt, p38, and JNK [149]. 

## 6. Conclusions

There is an ongoing need for new therapeutic modalities against SARS-CoV-2 infection that continues to spread rapidly around the world. Mounting evidence shows that hyper-inflammation is the hallmark of COVID-19 pathophysiology leading to significant morbidity and mortality. The majority of the histamine secreted by mast cells may play an important role in the pathophysiology of COVID-19 and is regarded as a promising pharmacological target. The activation of pulmonary mast cells releases mediators with proinflammatory, fibrotic, and thrombogenic properties. Moreover, observational studies have shown the potential benefits of H2 receptor antagonists in patients with COVID-19. However, given the relative paucity of agents targeting mast cells, it may be rational to consider alternative treatments with pleiotropic properties including the modulation of histamine release. Mast cell-derived histamine can regulate not only adaptive and immune system responses but also vasodilatation by binding to endothelial H1 receptors and enhancing NO production. In an inverse way, histamine-induced NO can negatively modulate mast cell activation, mediator expression, and secretion, thus creating an autocrine loop [152]. In this context, several in vivo and in vitro studies indicate that mast cell activity can be regulated by various nutraceuticals that have gained interest for the treatment of COVID-19. In this way, immunonutrition could lead to a reduction in the de novo synthesis and/or release of histamine and other mast cell mediators that are considered to mediate, at least in part, the immune and microvascular alterations present in COVID-19 (Figure 1). These regimens could be used prophylactically or adjunctively to the conventional treatment of patients infected with SARS-CoV-2. We should point out that for other nutrients, such as glutamine and arginine that have been extensively studied for their immune modifying effect, there are no data available regarding their role on mast cells and histamine during SARS-CoV-2 infection. Nevertheless, the clinical evidence is still limited, and further investigations are necessary to validate the efficacy of nutraceuticals in managing the immune response in COVID-19, and, in particular, modulating mast cell activity.

## Figures and Tables

**Figure 1 nutrients-13-03458-f001:**
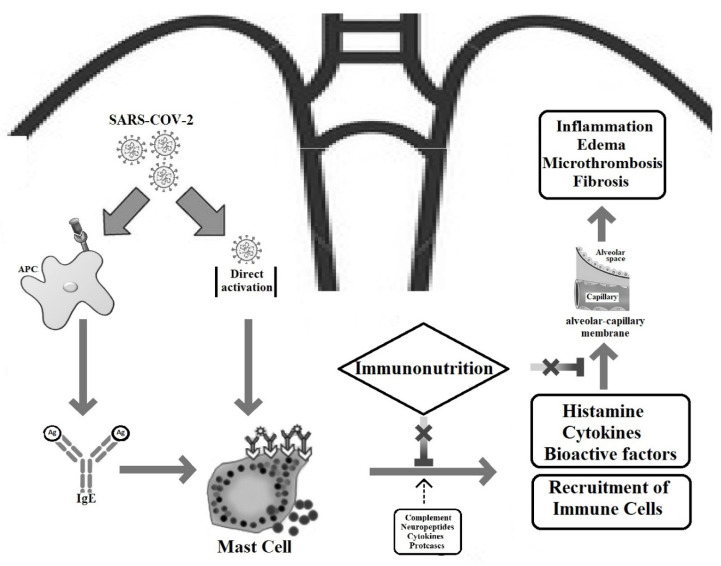
Schematic representation of the modulatory activity of immunonutrients with potential use in COVID-19 on mast cells and histamine during SARS-CoV-2 infection. APC: antigen-producing cells (macrophages or dendritic cells).

## Data Availability

Not applicable.

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
