# Peer review of "The Complex Interplay between Immunonutrition, Mast Cells, and Histamine Signaling in COVID-19"

_nutrients, 2021, doi:10.3390/nu13103458_

Round 1

Reviewer 1 Report

This is an interesting, well-conceived review manuscript on an important topic to understand the metabolic relationships and reciprocal influences between the COVID-19 disease and immunity from a nutrition point of view, before during, and after the infection. I do see a good point in publishing this narrative review for nutritionists, dietitians, and clinical doctors involved in the nutrition field.

Say that, please correct this mistyping and revise the manuscript following my suggestions, just to harmonize more the concepts on immuno-nutrition.

Mistyping:

S at lane 30 not s. Then entering at lane 31 not thenentering. Injury is considered the at lane 36 not injuryisconsideredthe. Hasimplicated and inCOVID-19 at line 63, SARS_COV-2 at lines 73, 187, 188, inhibitshistamine at line 102, acidsare at line 198 and so on along the text.

In the introduction section

“Severe lung injuryisconsideredthe  result of immune hyperreaction and a cytokine storm that trigger immune-mediated lung  inflammation and increased vascular permeability [6,8]. Endothelial injury and micro-thromboses ensue in the lungs and other organs of COVID-19 patients [9,10]. Patients may require mechanical ventilation and develop multiple organ failure [5,6].”

Please take off these sentences below and replace them with sentences related to immune system components. Please write a better explanation of innate and adaptive immunity. Mast cells cover both aspects.

Please read and add this reference: Role of mast cells in innate and adaptive immunity. Journal of biological regulators and homeostatic agents, 26(2), 193–201.

In the Vitamin D section  

Please add this reference: Immunological Response to SARS-CoV-2 Is Sustained by Vitamin D: A Case Presentation of One-Year Follow-Up. Reports 2021, 4, 18.

In the Selenium section

Add the concept of selenocysteine which is defined for a set of proteins having a role in prostanoids metabolism and therefore immunity.

Please add this reference: Selenium and selenoproteins in prostanoid metabolism and immunity. Critical reviews in biochemistry and molecular biology, 54(6), 484–516.

In the Flavonoids and Curcumin sections

Here the message for readers is not clear please specifically indicate that SARS-CoV here is referred to as the first beta-coronavirus infection.

Author Response

On behalf of all the authors, I would like to thank you very much for your valuable and thorough revision of our manuscript. Our answers and the changes in the text are marked in red. In the next lines, a point-by-point response to each comment is presented. Please, have in mind that line numbers are now changed due to the insertion of extra References and responses to Reviewers comments.

Reviewer #1

"This is an interesting, well-conceived review manuscript on an important topic to understand the metabolic relationships and reciprocal influences between the COVID-19 disease and immunity from a nutrition point of view, before during, and after the infection. I do see a good point in publishing this narrative review for nutritionists, dietitians, and clinical doctors involved in the nutrition field.

Say that, please correct this mistyping and revise the manuscript following my suggestions, just to harmonize more the concepts on immuno-nutrition.

Mistyping:

S at lane 30 not s. Then entering at lane 31 not then entering. Injury is considered the at lane 36 not injury is considered the. Has implicated and inCOVID-19 at line 63, SARS_COV-2 at lines 73, 187, 188, inhibits histamine at line 102, acids are at line 198 and so on along the text.

We thank the reviewer#1 for the comments. All mistakes regarding mistyping have now been corrected.

In the introduction section

“Severe lung injury is considered the result of immune hyperreaction and a cytokine storm that trigger immune-mediated lung inflammation and increased vascular permeability [6,8]. Endothelial injury and micro-thromboses ensue in the lungs and other organs of COVID-19 patients [9,10]. Patients may require mechanical ventilation and develop multiple organ failure [5,6].”

Please take off these sentences below and replace them with sentences related to immune system components. Please write a better explanation of innate and adaptive immunity. Mast cells cover both aspects.

Please read and add this reference: Role of mast cells in innate and adaptive immunity. Journal of biological regulators and homeostatic agents, 26(2), 193–201.

We thank the reviewer for the comment. We have now added the specific reference and changed text to Severe lung injury in SARS-COV-2 patients is considered the result of immune hyperreaction that involves both innate and adaptive immune responses. Briefly, coronavirus infection activates antigen-presenting cells, such as macrophages, that display viral antigens to T and B cells resulting in antibody production and increased cytokine secretion in the form of a cytokine storm. Other immune cells are also implicated, including mast cells that are important coordinators for both innate and adaptive immunity.” (lines 62-67 in the revised version)

In the Vitamin D section 

Please add this reference: Immunological Response to SARS-CoV-2 Is Sustained by Vitamin D: A Case Presentation of One-Year Follow-Up. Reports 2021, 4, 18.

We thank the reviewer for the comments. We have added the specific reference and revised our manuscript to «Recent data also indicate a positive effect of vitamin D supplementation on functional humoral immunity levels as determined by IgG levels.” (lines 151-152 in the revised version)

In the Selenium section

Add the concept of selenocysteine which is defined for a set of proteins having a role in prostanoids metabolism and therefore immunity.

Please add this reference: Selenium and selenoproteins in prostanoid metabolism and immunity. Critical reviews in biochemistry and molecular biology, 54(6), 484–516.

We thank the reviewer for the comments. We have added the reference. In the revised manuscript we revised text to “Published data also highlight the important role of biologic functions that occur  via incorporations of selenium into selenoproteins, in the form of selenocysteine amino acid residue. Selenocysteine (Sec-Cys)) is involved in a variety of prostanoid metabolism processes and  therefore have an impact in  immunity.” (lines 248-251 in the revised version)

In the Flavonoids and Curcumin sections

Here the message for readers is not clear please specifically indicate that SARS-CoV here is referred to as the first beta-coronavirus infection.

We thank the reviewer for the comment. We have now added the following revised sentences ”Curcumin may also exhibit antiviral activities and has been shown to hamper the replication and proliferation of SARS-CoV-1, the first beta-coronavirus that caused 2003 SARS and shares a substantial genetic similarity with SARS-CoV-2” (lines 321-324 in the revised version)

Reviewer 2 Report

Kekavas et al in this paper demonstrated the role of several immune-nutrients and its beneficial effect in COVID 19 pathology. The paper is well written, comprehensive and well-articulated. However, there are some minor revisions suggested.

Title: Histamine is released in addition to mast cells from a number of innate and adaptive immune cells as well as endothelial cells. So, if authors can avoid using the mast cell histamine it will more specifically address the scope of the article. Instead, authors can use The implication of immunonutrients, mast cell, and histamine signaling in covid 19 pathophysiology contexts, something along the line.

Immunonutrition is a relatively new terminology, if authors can broadly define it and why it is important in physiology and its implication in diseases that will orient readers for further reading of the paper.

Line 52, 53 : The release of histamine by perivascular vascular mast cells not only affects blood vasculature itself but also adjacent lymphatic vessel function inducing immune cell trafficking through its lumen. This can potentially contribute to acute inflammatory stimulus. (Pal et al, PMID : 31913658)

Line 56: Th1 Th2 polarization incorporation of this paper is strongly suggested from a histamine receptor perspective. Jutel et al, Nature PMID 11574888

Line 69: Mast cells have two subsets: chymase positive and tryptase positive. Which one is relevant in lung covid response?

Mast cell is shown to express histamine receptors by themselves which in an autocrine fashion can potentially ensue a feedback regulation further enhancing inflammatory response. Line 70, 71. Incorporation of this work is recommended. (Carlos et al, PMID 16703563, Pal et al, PMID: 31913658)

Line 145,146: reference needed to support this statement. Substitution of the word overreaction is suggested. Insights of the mechanism/s how vitamin c can be beneficial in covid 19 could bring a more well depth perspective.

Line 154, 155 references needed to support this statement

Line 175: Mechanism of how zinc regulates NFKb needed to be described a bit descriptive manner, In addition, activation of NFkB can also activate mast cell thereby releasing histamine secretion ensuing inflammatory response along with cytokine secretion ( Nizamutdinova et al, PMID 27875806, Pal et al 32625213)

Line 282: instead of mast cell histamine it will be better to use the majority of the histamine secreted by mast cells.  Line 286 -287: it would be better to give a context on how the mast cell-histamine loop not only regulate adaptive and immune system responses but also vascular physiology. ( Galli et al, PMID 32340580 )

Schematic: if the signaling cascades or events can be pointed out by some arrow direction that will create more clarity. Also, the schematic representation can be improved so that it can be scientifically and aesthetically better represented.

Author Response

Kekavas et al in this paper demonstrated the role of several immune-nutrients and its beneficial effect in COVID 19 pathology. The paper is well written, comprehensive and well-articulated. However, there are some minor revisions suggested.

We appreciate and thank the Reviewer#2 for the statement that our paper is interesting. We hope that our responses will help to clarify the concerns of the Reviewer.

Title: Histamine is released in addition to mast cells from a number of innate and adaptive immune cells as well as endothelial cells. So, if authors can avoid using the mast cell histamine it will more specifically address the scope of the article. Instead, authors can use The implication of immunonutrients, mast cell, and histamine signaling in covid 19 pathophysiology contexts, something along the line.

We thank the reviewer for the useful comments. The title of our manuscript changed according to your suggestion “The complex interplay between immunonutrition, mast cell, and histamine signaling in Covid 19“

Immunonutrition is a relatively new terminology, if authors can broadly define it and why it is important in physiology and its implication in diseases that will orient readers for further reading of the paper.

We thank the reviewer for the useful comments. Please see lines 114-118 in the revised manuscript “while a number of recently published key studies suggest promising effects of immunonutrition on acute respiratory infections. Briefly, immunonutrition can be defined as modulation of either the activity of the immune system, or modulation of the consequences of activation of the immune system, by nutrients or specific food items fed in amounts above those normally encountered in the diet.”

Line 52, 53: The release of histamine by perivascular vascular mast cells not only affects blood vasculature itself but also adjacent lymphatic vessel function inducing immune cell trafficking through its lumen. This can potentially contribute to acute inflammatory stimulus. (Pal et al, PMID : 31913658) (Pal et al, PMID : 31913658)

We have now added the suggested references. Please see lines 81-85 in the revised manuscript. “It also activates acute immune-mediated reactions, enhances vascular smooth muscle contraction and migration of other immune cells, antibodies, and mediators into the site of insult. The release of histamine by perivascular vascular mast cells also adjacent lymphatic vessel function inducing immune cell trafficking through its lumen, which potentially contribute to acute inflammatory stimulus”

Line 56: Th1 Th2 polarization incorporation of this paper is strongly suggested from a histamine receptor perspective. Jutel et al, Nature PMID 11574888

We thank the reviewer for the useful comment. We have now added the suggested reference (Jutel, M.; Watanabe, T.; Klunker, S.; Akdis, M.; Thomet, O.A.; Malolepszy, J.; Zak-Nejmark, T.; Koga, R.; Kobayashi, T.; Blaser, K., et al. Histamine regulates T-cell and antibody responses by differential expression of H1 and H2 receptors. Nature 2001, 413, 420-425, doi:10.1038/35096564.)

Line 69: Mast cells have two subsets: chymase positive and tryptase positive. Which one is relevant in lung covid response?

We thank the reviewer for the useful comment. Please see lines 96-97 in the revised manuscript “In animal models of COVID-19, mast cells detected in the lungs were chymase positive”

Mast cell is shown to express histamine receptors by themselves which in an autocrine fashion can potentially ensue a feedback regulation further enhancing inflammatory response. Line 70, 71. Incorporation of this work is recommended. (Carlos et al, PMID 16703563, Pal et al, PMID: 31913658)

We thank the reviewer for the useful comment. We have now added the suggested references and modified text to “Mast cell is shown to express histamine receptors by themselves which in an autocrine fashion can potentially ensue a feedback regulation further enhancing inflammatory response (Carlos et al, PMID 16703563, Pal et al, PMID: 31913658)” (lines 97-99 in the revised version)”

Line 145,146: reference needed to support this statement. Substitution of the word overreaction is suggested. Insights of the mechanism/s how vitamin c can be beneficial in covid 19 could bring a more well depth perspective.

We have now added an extra reference (Ellulu M.S., Rahmat A., Patimah I., Khaza'ai H., Abed Y. Effect of vitamin C on inflammation and metabolic markers in hypertensive and/or diabetic obese adults: A randomized controlled trial. Drug Des Devel Ther. 2015;9:3405–3412) and changed text to “Previous studies have shown that vitamin C administration attenuates robust immune response” (lines 189-190 in the revised version)”

Regarding the mechanisms, please see lines XXXX in the revised manuscript “A daily oral dosage of 1–2 g/day of vitamin C has been proposed as beneficial for the prevention or treatment of COVID-19, while higher doses of intravenous vitamin C, up to 24 gm/day, are being evaluated in critically ill patients with COVID-19. Proposed mechanisms for the ability of vitamin C to benefit patients with COVID-19  point to the prevention of IL-6 increase of in several (pro)inflammatory conditions  and the inhibition to the increase of a range of inflammatory cytokines”  (lines 186-188 in the revised version)

Line 154, 155 references needed to support this statement

We thank the reviewer for the useful comment, and we have now added the following reference:

Maintz & Novak. Am J Clin Nutr 2007; 1185-96) (lines 199 in the revised version)

Line 175: Mechanism of how zinc regulates NFKb needed to be described a bit descriptive manner, In addition, activation of NFkB can also activate mast cell thereby releasing histamine secretion ensuing inflammatory response along with cytokine secretion ( Nizamutdinova et al, PMID 27875806, Pal et al 32625213)

We thank the reviewer for the useful comments. Please see lines 227-232 in the revised manuscript:

“Various mechanisms have been suggested, but modulation of the NF-kB pathway could be the result of inhibition of cyclic nucleotide phosphodiesterase, cross activation of protein kinase A and inhibitory phosphorylation of protein kinase Raf-1 In addition, activation of NF-kB can also activate mast cell thereby releasing histamine secretion ensuing inflammatory response along with cytokine secretion.

Line 282: instead of mast cell histamine it will be better to use the majority of the histamine secreted by mast cells. Line 286 -287: it would be better to give a context on how the mast cell-histamine loop not only regulate adaptive and immune system responses but also vascular physiology. (Galli et al, PMID 32340580 )

We thank the reviewer for the useful comments. Please see lines 339-340 in the revised manuscript “The majority of the histamine secreted by mast cells” and lines 345-349 ” Mast cell-derived histamine can regulate not only adaptive and immune system responses, but also vasodilatation by binding to endothelial H1-receptors and enhancing NO production. In an inverse way, histamine induced NO can negatively modulate mast cell activation, mediator expression and secretion, thus creating an autocrine loop”

Schematic: if the signaling cascades or events can be pointed out by some arrow direction that will create more clarity. Also, the schematic representation can be improved so that it can be scientifically and aesthetically better represented.

We thank the reviewer for the useful comments. Please see the update figure in the revised version of our manuscript
